# Neural MMO 2.0: A Massively Multi-task Addition to Massively Multi-agent Learning

**Joseph Suárez**                                JSUAREZ@MIT.EDU

**Phillip Isola**                                PHILLIPI@MIT.EDU

*Massachusetts Institute of Technology*

**Kyoung Whan Choe**                             CHOE.KYOUNG@GMAIL.COM

**David Bloomin**                                DAVEEY@GMAIL.COM

**Hao Xiang Li**                                 HXL23@CAM.AC.UK

**Nikhil Pinnaparaju**                           NIKHILPINNAPARAJU@GMAIL.COM

**Nishaanth Kanna**                              NISHAANTHKANNA@GMAIL.COM

**Daniel Scott**                                 DSCOTT45@GATECH.EDU

**Ryan Sullivan**                                RSULLI@UMD.EDU

**Rose S. Shuman**                               ROSE.SHUMAN@ALUMNI.BROWN.EDU

**Lucas de Alcântara**                           LUCASAGLLEITE@GMAIL.COM

**Herbie Bradley**                               HB574@CAM.AC.UK

**Louis Castricato**                             LOUIS_CASTRICATO@BROWN.EDU

*CarperAI*

**Kirsty You**                                   KIRSTYYOU@CHAOCANSHU.AI

**Yuhao Jiang**                                  YUHAOJIANG@CHAOCANSHU.AI

**Qimai Li**                                     QIMAILI@CHAOCANSHU.AI

**Jiaxin Chen**                                  JIAXINCHEN@CHAOCANSHU.AI

**Xiaolong Zhu**                                 XIAOLONGZHU@CHAOCANSHU.AI

*Parametrix.AI*

## Abstract

Neural MMO 2.0 is a massively multi-agent environment for reinforcement learning research. The key feature of this new version is a flexible task system that allows users to define a broad range of objectives and reward signals. We challenge researchers to train agents capable of generalizing to tasks, maps, and opponents never seen during training. Neural MMO features procedurally generated maps with 128 agents in the standard setting and support for up to. Version 2.0 is a complete rewrite of its predecessor with three-fold improved performance and compatibility with CleanRL. We release the platform as free and open-source software with comprehensive documentation available at neuralmmo.github.io and an active community Discord. To spark initial research on this new platform, we are concurrently running a competition at NeurIPS 2023.

37th Conference on Neural Information Processing Systems (NeurIPS 2023) Track on Datasets and Benchmarks.

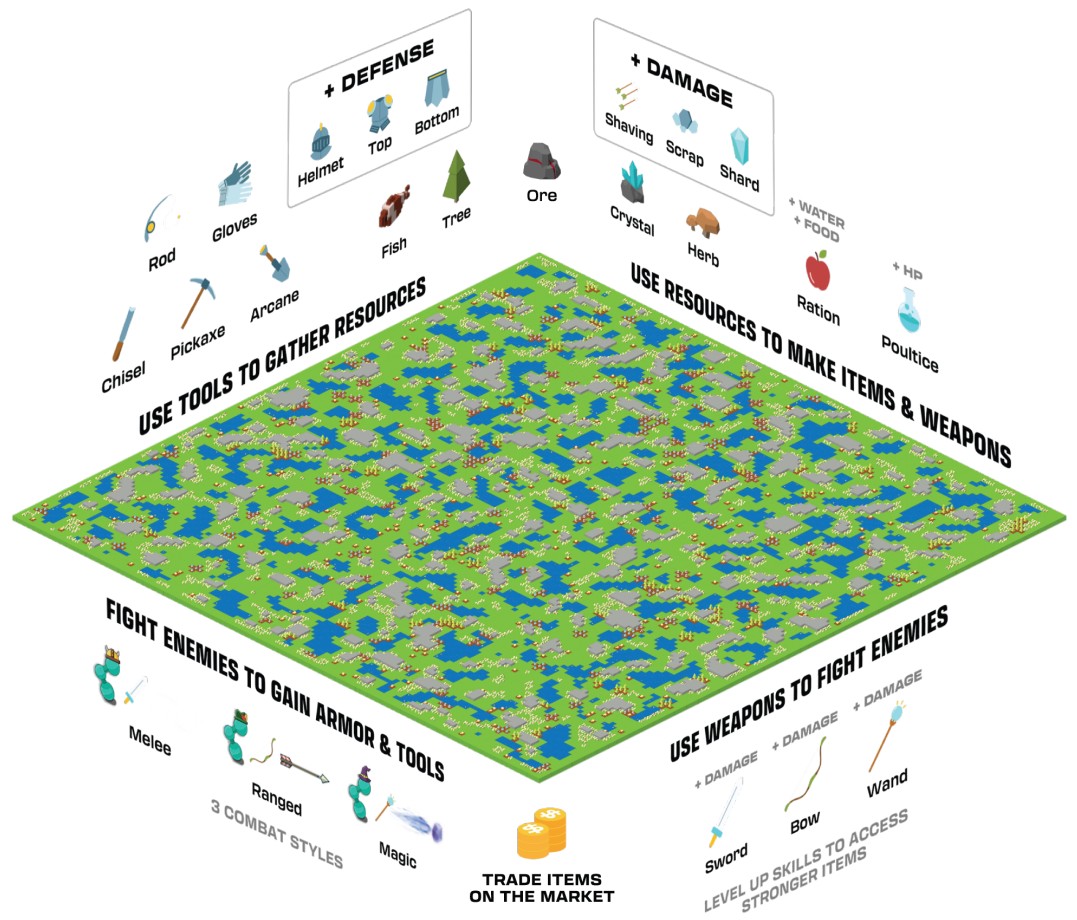

Figure 1: Overview of Neural MMO 2.0. Users can define tasks to specify a broad range of agent objective. In general, these involve using tools to gather resources, using resources to make items and weapons, using weapons to fight enemies, and fighting enemies to gain armor and tools. Full documentation is available at neuralmmo.github.io.

## 1 Novelty and Impact

Neural MMO is a reinforcement learning platform first released in 2019 [Suarez et al., 2019], with updates featured in short-form at AAMAS 2020 [Suarez et al., 2020] and ICML 2020, and a new version published in the 2021 NeurIPS Datasets & Benchmarks track [Suarez et al., 2021]. Since then, the platform has gained traction through competitions at IJCAI 2022 and NeurIPS 2022, totaling 3500+ submission from 1200+ users, which significantly improved state-of-the-art on the platform. Alongside these developments, our community on Discord has grown to nearly 1000 members.

While previous versions of the environment defined fixed objectives through only the reward signal, Neural MMO 2.0 introduces a flexible task system that allows users to define per-agent or per-team objectives and rewards, expanding the platform's applicability to a broader range of problems. In particular, Neural MMO 2.0 enables research on generalization, open-endedness, and curriculum learning—areas that were difficult to explore with prior versions and which require sophisticated, flexible simulators. There are few if any other environments of comparable scope to Neural MMO available for these problems.

## Neural MMO Gameplay, Features, and Mechanics

**Items in Inventory**

Items
7    16

**Weapons in Inventory**

Items
19

**Map Features**

Attacker
Water
Runes
Ranged Attack
Food
Arrows
Rations
Target

Health
Food
Water

**NPCs**

Agent    Warrior    Mage    Archer

**Gather Ore → Level Up Prospecting**

| Level | | | |
|---|---|---|---|
| Melee | 0 | Prospecting | 2 |
| Range | 0 | Caving | 0 |
| Mage | 0 | Alchemy | 0 |
| Fishing | 0 | Herbalism | 0 |

**Exchange Market**

| Market | | | | | | | | | | | | | | |
|---|---|---|---|---|---|---|---|---|---|---|---|---|---|---|
| Bow | Level | Price | Wand | Level | Price | Spear | Level | Price | Ration | Level | Price | | | |
| | Lv 1 | - | | Lv 1 | - | | Lv 1 | - | | Lv 1 | - | | | |
| | Lv 2 | - | | Lv 2 | - | | Lv 2 | - | | Lv 2 | - | | | |
| | Lv 3 | - | | Lv 3 | - | | Lv 3 | - | | Lv 3 | - | | | |
| | Lv 4 | - | | Lv 4 | - | | Lv 4 | - | | Lv 4 | - | | | |
| | Lv 5 | - | | Lv 5 | - | | Lv 5 | - | | Lv 5 | - | | | |
| | Lv 6 | - | | Lv 6 | - | | Lv 6 | - | | Lv 6 | - | | | |
| | Lv 7 | - | | Lv 7 | - | | Lv 7 | - | | Lv 7 | - | | | |
| | Lv 8 | - | | Lv 8 | - | | Lv 8 | - | | Lv 8 | - | | | |
| | Lv 9 | - | | Lv 9 | - | | Lv 9 | - | | Lv 9 | - | | | |
| | Lv 10 | - | | Lv 10 | - | | Lv 10 | - | | Lv 10 | - | | | |
| Whetstone | Level | Price | Rune | Level | Price | Arrow | Level | Price | Potion | Level | Price | | | |
| | Lv 1 | - | | Lv 1 | - | | Lv 1 | - | | Lv 1 | - | | | |
| | Lv 2 | - | | Lv 2 | - | | Lv 2 | - | | Lv 2 | - | | | |

Figure 2: Neural MMO 2.0 features procedurally generated terrain, 7 resources to collect, 3 combat styles, 5 gathering and 3 combat professions to train and level up, scripted NPCs that roam the map, and 16 types of items in 10 quality levels including weapons, armor, consumables, tools, and ammunition. An environment-wide market allows agents to trade items with each other.

Practical engineering improvements are at the core of Neural MMO 2.0. These include:

1. A 3x faster engine. This was developed as part of a complete rewrite of our 5+ year old code base and is particularly important for reinforcement learning research, where simulation is often the bottleneck. For example, the upcoming competition would not be practical on the old engine.

2. Simple baselines with CleanRL, a popular and user-friendly reinforcement learning library. CleanRL and most other reinforcement learning frameworks are not natively compatible with environments of this complexity, and previous versions required convoluted, environment-specific compatibility wrappers. Neural MMO 2.0 integrates PufferLib to solve this problem.

3. A web client available at neuralmmo.github.io/client, generously open-sourced by Parametrix.AI. This client offers improved visualization capabilities and eliminates setup requirements.

Additionally, the platform's documentation has been professionally rewritten in consultation with the development team. This, along with a more intuitive and accessible website layout, marks a significant step towards improving user engagement. A collection of papers detailing previous versions and competitions is available on neuralmmo.github.io.

## 2 Neural MMO 2.0

Neural MMO (NMMO) is an open-source research platform that is computationally accessible. It enables populations of agents to be simulated in procedurally generated virtual worlds. Each world features unique landscapes, non-playable characters (NPCs), and resources that change each round. The platform draws inspiration from Massively Multiplayer Online games (MMOs), which are online video games that facilitate interaction among a large number of players. NMMO is a platform for intelligent agent creation, typically parameterized by a neural network. Agents in teams must forage for resources to stay alive, mine materials to increase their combat and task completion capabilities, level up their fighting styles and equipment, practice different professions, and engage in trade based on market demand.

In the canonical setting of NMMO that will support the upcoming competition, users control 8 out of a total of 128 simulated agents. The ultimate goal is to score more points by completing more tasks than the other 118 agents present in the same environment. Originally, we planned to introduce team-based tasks and objectives, but we decided to postpone the introduction of these given the practical limitations of learning libraries. After the conclusion of the competition, top submissions will be provided as baseline opponents. NMMO includes the following mechanisms to induce complexity into the environment:

- Terrain: Navigate procedurally generated maps
- Survival: Forage for food and water to maintain your health
- NPC: Interact with Non-Playable Characters of varying friendliness
- Combat: Fight other agents and NPCs with Melee, Range, and Magic
- Profession: Use tools to practice Herbalism, Fishing, Prospecting, Carving, and Alchemy
- Item: Acquire consumables and ammunition through professions
- Equipment: Increase offensive and defensive capabilities with weapons and armor
- Progression: Train combat and profession skills to access higher level items and equipment
- Exchange: Trade items and equipment with other agents on a global market

A detailed wiki is available on the project's document site.

## 3   Background and Related Work

In the initial development phase of Neural MMO from 2017 to 2021, the reinforcement learning community witnessed the release of numerous influential environments and platforms. Particularly noteworthy among these are Griddly [Bamford et al., 2020], NetHack [Küttler et al., 2020], and MineRL [Guss et al., 2021]. A comprehensive comparison of these with the initial Neural MMO can be found in our previous publication [Suarez et al., 2021]. The present work primarily focuses on recent advancements in the reinforcement learning environments sphere. Griddly has sustained ongoing enhancements, while MineRL has inspired several competitive initiatives. Since 2021, only a few new environments have emerged, with the most pertinent ones being Melting Pot [Leibo et al., 2021], and XLand [Team et al., 2021]. Melting Pot and its successor, Melting Pot 2.0 [Agapiou et al., 2023], comprise many multiagent scenarios intended for evaluating specific facets of learning and intelligence. XLand and its sequel, XLand 2.0 [Team et al., 2023], present large-scale projects focusing on training across a varied curriculum of tasks within a procedurally generated environment, with a subsequent emphasis on generalization to novel tasks. Compared to Melting Pot, Neural MMO is a larger environment with flexible task specifications, as opposed to a set of individual scenarios. XLand, while architecturally akin to Neural MMO, predominantly explores two-agent settings, whereas Neural MMO typically accommodates 128. A crucial distinction is that XLand is primarily a research contribution enabling the specific experiments presented in the publication. It does not provide open-source access and is not computationally practical for academic-scale research. Conversely, Neural MMO is an open-source platform designed for computational efficiency and user-friendliness.

## 4   Task System

The task system of Neural MMO 2.0, a central component of the new version, comprises three interconnected modules: GameState, Predicates, and Tasks. This system leverages the new Neural MMO engine to provide full access to the game state in a structured and computationally efficient manner. This architectural enhancement surpasses the capabilities of Neural MMO 1.x, allowing users to precisely specify the tasks for agents, paving the way for task-conditional learning and testing generalization to unseen tasks during training.

### 4.1   GameState

The GameState module acts as a high-performance data manager, hosting the entire game state in a flattened tensor format instead of traditional object hierarchies. This vectorization serves a dual

purpose: first, it accelerates simulation speeds—a crucial factor in generating data for reinforcement learning; and second, it offers researchers an efficient tool to cherry-pick the required bits of data for defining objectives. While this format was originally inspired by the data storage patterns used in MMOs, adaptations were needed to support the computation of observations and definition of tasks.

Alongside GameState, we also introduced auxiliary datastores to capture *event* data—unique in-game occurrences that would be not be captured otherwise. These datastores record things that happen, such as when an agent lands a successful hit on an opponent or gathers a resource, rather than just the outcomes, i.e. damage inflicted or a change in tile state. Events enable the task system to encompass a broader range of objectives in a computationally efficient manner.

To illustrate the flexibility provided by GameState access, let's walk through some representative query examples. The snippets in the GameState Appendix employ both the global and agent-specific GameState queries. Global access is useful for game dynamics such as time and environmental constants. We also provide a convenience wrapper for accessing agent-specific data.

This query API gives researchers direct access to the mechanics of the game environment, offering a rich playground for studying complex multi-agent interactions, resource management strategies, and competitive and cooperative dynamics in a reinforcement learning context.

### 4.2 Predicates

The Predicates module offers a robust syntax for defining completion conditions within the Neural MMO environment.

Predicates interface with the game state (the "subject") to provide convenient access to agent data and any additional arguments desired. Predicates return a float ranging from 0 to 1, rather than a boolean. This design choice supports partial completion of predicates—crucial for generating dense reward functions—while still allowing tasks to be considered complete when the return value equals 1. As a starting point, Neural MMO offers 25 built-in predicates that can access every aspect of NMMO. The first example in the Predicates Appendix illustrates the creation of a more complex objective, building on the game state and subject from the previous section.

The second example in the Predicates Appendix demonstrates how the Predicate system can be used to articulate complex, high-level objectives. The *FullyArmed* predicate demands that a specific number of agents in a team be thoroughly equipped. An agent is considered fully equipped if it has an entire set of equipment (hat, top, bottom, weapon, ammo) of a given level. To acquire a complete equipment set, agents would need to utilize various professions in different locations on the game map, which could take several minutes to accomplish. This task's complexity could be further amplified by setting a condition that each team member be outfitted specifically with melee, ranged, or magical equipment, necessitating the coordinated use of all eight professions.

### 4.3 Tasks

The Task API allows users to formulate tasks by combining predicates and assigning per-agent rewards based on the outcomes of intermediary predicates. This approach not only maintains an account of tasks completed but also provides a denser reward signal during training. We expect that most users will form tasks using the library of pre-built predicates. For advanced users, direct access to GameState enables mapping conditions on the game's internal variables to rewards, circumventing the need for intermediate predicates. The predicate can then be turned into a task. See the Tasks Appendix for an example.

## 5 Performance and Baselines

Neural MMO 2.0's new engine runs at approximately 3,000 agent steps per CPU core per second, up from the approximately 800 to 1,000 in the previous version. Its design focuses on native compatibility with a vectorized datastore that represents game state. This allows us to keep the environment in Python while maintaining efficiency, providing easier access for researchers looking to modify or extend Neural MMO.

Simulation throughput is highly dependent upon agent actions within the game. We compute statistics by having agents take random actions, but to maintain a fair estimate, we eliminate mortality since dead agents do not require any computation time. Given that NMMO equates one action to 0.6 seconds of real time, a single modern CPU core can simulate at 5,000 times real-time per-agent, equivalent to 250M agent steps or roughly 2.5 terabytes of data per day at approximately 10 KB per observation.

We also release a baseline model with training code and pretrained checkpoints. Compared to the previous TorchBeast [Küttler et al., 2019] baseline, our new model builds on top of CleanRL. This is a simpler library that is much easier to work with, but it is not designed to work with complex environments like Neural MMO by default. To achieve interoperability, we integrate with PufferLib, a library designed to streamline the various complexities of working with sophisticated environments.

## 6    Limitations

Despite its enhancements, Neural MMO 2.0 does not incorporate any novel game mechanics absent in version 1.x. However, in the most recent competition, even the top approaches did not learn to comprehend and utilize all of the game systems, and there is substantial room for improvement. Moreover, agent specialization within a team remained limited. These circumstances are likely attributable to the overly broad survival objective that invariably promotes dominant strategies, posing a challenge to balance. However, with the introduction of a more flexible task system in Neural MMO 2.0, we redefine performance as the capability to execute novel tasks, thereby enabling researchers to harness the existing game mechanics in a way not feasible in earlier versions.

## 7    Accessibility and Accountability

Neural MMO has been under active development with continuous support for the past 6 years. Each of the six major releases in this period was accompanied by comprehensive documentation updates, a guarantee of timely user support, and direct access to the development team via through the community Discord. The project will continue to support and maintenance. A fourth competition has been accepted to NeurIPS 2023 and is expected to improve the current baseline. The code for this project is hosted in perpetuity by the Neural MMO GitHub organization under the MIT license. We provide both a pip package and a containerized setup including the baselines. Documentation is consistently available on neuralmmo.github.io with no major outages recorded to date. The entire project is available as free and open-source software under the MIT license.

Neural MMO implements the standard PettingZoo [Terry et al., 2021] ParallelEnv API, a direct generalization of the OpenAI Gym [Brockman et al., 2016] API for multi-agent environments. Our baselines utilize CleanRL's [Huang et al., 2021] Proximal Policy Optimization (PPO) [Schulman et al., 2017] implementation, one of the simplest and most widely used reinforcement learning frameworks, with all algorithmic details encapsulated in a single file of approximately 400 lines. While CleanRL was originally designed for simpler environments like single-agent Atari [Bellemare et al., 2012] games, Neural MMO extends its capabilities through PufferLib, which provides native compatibility through a multiagent vectorization backend. The details of this library are available at pufferai.github.io.

## 8    Ethics and Responsible Use

Neural MMO is an abstract game simulation featuring systems of combat and commerce. These elements are incorporated for visual interpretability and are not representative of any actual violence or commerce systems. We are confident that these systems are sufficiently removed from their real-world counterparts that Neural MMO would not be a useful training platform for developing such systems. The use of game-like elements in Neural MMO is a deliberate choice to align with human intuition and does not reflect any specific real-world scenario. Neural MMO's primary objective is to facilitate research on understanding and advancing the capabilities of learning agents. The project does not include any real-world human data other than the code and documentation voluntarily submitted by contributors and some 3D asset files commissioned at fair market rate.

## 9  Conclusion

Neural MMO 2.0 is a significant evolution of the platform. We invite researchers to tackle a new challenge in generalization across unseen new tasks, maps, and adversaries. Furthermore, we have achieved significant advancements in computational efficiency, yielding a performance improvement of over 300%, and have ensured compatibility with popular reinforcement learning frameworks like CleanRL. This opens up the potential for broader utilization by researchers and makes the environment significantly more accessible, especially to those working with more modest computational resources. Neural MMO has a five-year history of continuous support and development, and we commit to maintaining this support, making necessary adaptations, and facilitating a lively and active community of users and contributors. With the concurrent NeurIPS 2023 competition, we look forward to sparking new research ideas, encouraging scientific exploration, and contributing to progress in multi-agent reinforcement learning.

## Acknowledgements

Training compute for baselines provided by Stability AI, Carper AI, and Eleuther AI. Development for 2.0 was an open-source project under CarperAI led by Joseph Suarez and managed by Louis Castricato. Web client by Parametrix.AI with artwork by Lucas de Alcântara. Technical documentation by Rose S. Shuman in collaboration with the development team. Engine work for 2.0 by David Bloomin. Special thanks to Kyoung Whan Choe for major contributions to development and ongoing environment support. Original project by Joseph Suarez.

This work was supported in part by ONR MURI grant N00014-22-1-2740.

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

# A  Game State

```python
# True if any agent in subject can see a tile of tile_type
any(tile_type.index in t for t in subject.obs.tile.material_id)

# True if all subjects are alive.
np.count_nonzero(subject.health > 0) == len(subject)

# Computes the maximum l-inf distance between teammates
current_dist = max(subject.row.max()-subject.row.min(),
     subject.col.max()-subject.col.min())

# Tracks hits scored with a specific combat style
hits = subject.event.SCORE_HIT.combat_style == combat_style.SKILL_ID

# Computes the summed gold of all teammates
subject.gold.sum()

# Evalutaes to >= 1 if the current game tick is >= the specified tick
gs.current_tick / num_tick
```

# B Predicates

## B.1 Example 1

```python
# Signature for predicates
def predicate(gs: GameState, subject: Group, **kwargs) -> float:

def DistanceTraveled(gs: GameState, subject: Group, dist: int):
  """True if the summed l-inf distance between each agent's current pos and
  spawn pos is greater than or equal to the specified _dist."""
  if not any(subject.health > 0):
    return 0
  r, c = subject.row, subject.col
  dists = utils.linf(list(zip(r,c)),[gs.spawn_pos[id_]
        for id_ in subject.entity.id])
  return max(min(dists.sum() / dist, 1.0), 0.0)
```

## B.2 Example 2

```python
def FullyArmed(gs: GameState, subject: Group,
               combat_style: nmmo_skill.CombatSkill,
               level: int,  num_agent: int):
  """True if the number of fully equipped agents >= _num_agent

  To determine fully equipped, we compare the levels of
  the hat, top, bottom, weapon, ammo to _level."""
  WEAPON_IDS = {
    nmmo_skill.Melee: {'weapon':5, 'ammo':13}, # Spear, Whetstone
    nmmo_skill.Range: {'weapon':6, 'ammo':14}, # Bow, Arrows
    nmmo_skill.Mage: {'weapon':7, 'ammo':15} # Wand, Runes
  }
  item_ids = { 'hat':2, 'top':3, 'bottom':4 }
  item_ids.update(WEAPON_IDS[combat_style])
  lvl_flt = (subject.item.level >= level) & (subject.item.equipped > 0)
  type_flt = np.isin(subject.item.type_id,list(item_ids.values()))
  _, equipment_numbers = np.unique(
        subject.item.owner_id[lvl_flt & type_flt], return_counts=True)
  if num_agent == 0:
    return 1.0
  return max(min((equipment_numbers >= len(
        item_ids.items())).sum() / num_agent, 1.0), 0.0)
```

# C Tasks

```python
def KillPredicate(gs: GameState, subject: Group):
  """The progress, the max of which is 1, should
      * increase small for each player kill
      * increase big for the 1st and 3rd kills
      * reach 1 with 10 kills"""
  num_kills = len(subject.event.PLAYER_KILL)
  progress = num_kills * 0.06
  if num_kills >= 1:
    progress += .1
  if num_kills >= 3:
    progress += .3
  return min(progress, 1.0)
```

This predicate can be turned into a task like this.

```python
from nmmo.task import predicate_api
from nmmo.task.group import Group

# Create a Predicate class with  interfaces to GameState and Group
pred_cls = predicate_api.make_predicate(KillPredicate)

# Create a task for each agent
task_list = []
for agent_id in agent_list:
  task_list.append(pred_cls(subject=Group(agent_id)).create_task())

# Create a task that evaluates and rewards a whole team together
team_task = pred_cls(subject=Group(agent_list)).create_task()

# Make a task that agent 1 gets rewarded for the agent 2's evaluation
pred_cls = predicate_api.make_predicate(AllDead)
task_for_agent_1 = pred_cls(
        subject=Group(agent_2)).create_task(assignee=agent_1)
```

