# Supplementary Material for Neural MMO 2.0

August 19, 2023

## 1 Documentation and Intended Uses

Documentation is available at neuralmmo.github.io. Neural MMO 2.0 is a large-scale but computationally efficient environment with a flexible task structure. It is intended to promote agent-based intelligence research in complex, cognitively interesting simulations.

## 2 Author Statement

The lead author bears all responsibility in case of violation of rights etc, though none such are applicable because Neural MMO 2.0 is a simulated environment, not a real-world dataset. All code, models, and other such artifacts are MIT licensed.

## 3 Hosting, Licensing, and Maintenance plan

Code and docs hosted by GitHub, anticipated updates to be licensed under MIT or similarly permissive terms. This project has a 6-year development history and a 5-year continuous maintenance and support history. Development will continue for at least the next year with support and maintenance orchestrated via our community Discord during and thereafter.