# OpenReview forum: "Neural MMO 2.0: A Massively Multi-task Addition to Massively Multi-agent Learning"
_NeurIPS.cc/2023/Track/Datasets_and_Benchmarks — NeurIPS 2023 Datasets and Benchmarks Poster_

### Official Review · Reviewer_3mZR · 2023-07-21
**Review of NeurIPS 2023 Track Datasets and Benchmarks Submission68 from Reviewer 3mZR**

**Rating:** 6
**Confidence:** 4
**Correctness:** The code for the environment is well-…
**Clarity:** The paper is well-written and easy to…

**Strengths:**

- Version 2.0 added a user-friendly web UI allowing users to visualize the learning policies, which I think is a big improvement from the Unity-based visualization client from the older version. The web UI allows users to monitor the agents' behaviors frame-by-frame, e.g. market transactions, item consumptions, switching equipment, etc.
- The introduction of the market exchange system in version 2.0 broadened the trading capabilities of agents, permitting both intra-team and inter-team transactions.
- Version 2.0 further expanded on the Profession Progression mechanics, and intertwining it with the market exchange system. As agents start to specialize in diverse professions, their preferences for specific goods correspondingly rise, as do their acceptable price points for these goods. This intriguingly mirrors the economic law of supply and demand.
- The code is highly modular and easy to modify to fit new research needs.

**Additional Feedback:**

***Suggestions regarding the competitions***:

As I recollect that the top winning solutions from the top teams were pretty similar, i.e. MAPPO + FSP/PRSO + complex feature engineering, and most of these top teams are from well-funded large industrial labs (IIRC they are from NetEase Game, Bilibili Game, Tencent AI, Alibaba, etc.). It has been demonstrated that the path to success in this competition is not necessarily through innovating new algorithms, but rather by accumulating substantial computational resources to execute MAPPO in a massively parallel manner with large batch sizes (e.g. clusters with thousands of CPU cores and V100 GPUs). The $500 AWS compute credit offered to those in need falls short of bridging the gap between individual participants and industrial labs. While the organizers' decision to open-source the top solutions is commendable, it still presents a significant hurdle for the average user to reproduce these results. Replication involves not only understanding the methods used, but also access to equivalent computational resources, which may be out of reach for many. I acknowledge that this is not an issue with your environment, but is primarily due to the low sample efficiency of multi-agent RL algorithms. However, the realization that the resource disadvantage makes it virtually impossible for individual academic participants to compete successfully is disheartening. This could potentially harm the growth of the NeuralMMO community in the long run. Indeed, I recognize that the organizers have attempted to level the playing field by creating a separate track for offline RL. However, when it comes to final evaluation, offline RL approaches inevitably underperform when compared to online-trained algorithms like MAPPO + FSP. This effectively still creates an uneven playing field, with resources continuing to be a deciding factor.

I apologize if this seems like a personal complaint, but I'm genuinely curious if you've considered potential solutions to this issue. Perhaps creating separate tracks for academic researchers and industrial labs could be a viable approach?

***Other questions:***
- While my experience with this specific environment has been limited since the NeurIPS competition, I'm interested in any enhancements you may have implemented regarding intra-team trading mechanisms. Previously, all transactions were managed via a public market exchange system. My understanding is that the market order was executed sequentially based on agent ID, lacking a mechanism to verify if an agent was making purchases from its teammates. This system essentially bestowed priority only to those agents with an ID smaller than at least one of their teammates, allowing them to secure items at a highly competitive price before other agents could exploit this opportunity. While if your ID is the smallest of your team, then good luck finding any cheap items. Could you provide an update on this aspect of the system?

**Documentation:**

The documentation has sufficient details; Discord community support is good, and I can usually get the author's response within a day.

**Ethics:**

No obvious ethical risks in my opinions.

**Limitations:**

Ethical concerns have been adequately discussed in Section 4. Two paragraphs are dedicated to addressing the limitations of this environment, and there's the promise for the long-term support of this environment.

**Opportunities For Improvement:**

- One of my primary concerns revolves around the nature of the added features. While I commend the consistent efforts of the authors to enhance the environment, these changes, pardon my harsh words, appear to refine the existing product rather than revolutionize it fundamentally. The previous version, which this track accepted, was largely praised for its procedural generation of large-scale multi-agent maps and the concept of an MMO-style multi-agent environment. The features in v2.0 provide additional depth to the environment, transitioning it closer to a full-fledged video game.
- Presently, this environment poses a significant challenge for any multi-agent reinforcement learning algorithm to manage in an end-to-end manner. Many mechanisms, as I recall from the last competition, still rely heavily on scripting.
- Transition from RLlib to CleanRL, in my personal view, feels like a step backward considering the necessity for mass parallelism to train meaningful policies in this environment. Have you considered adding baselines that merge self-play algorithms (e.g. NFSP, FSP, PRSO-NASH), some of which were critical components of top-ranking solutions from previous competitions?

Nitpicking:
- Line 58: in-line citation shown as a question mark.

**Relation To Prior Work:**

Discussion of previous multi-agent environments is somewhat limited. A detailed survey presented in an organized format (e.g. table) will be much appreciated.

**Summary And Contributions:**

Neural MMO 2.0" is an open-source platform for reinforcement learning research that provides a multi-agent, multi-task environment with new task systems, training settings, and increased performance. The platform features procedural maps supporting 1-1024 agents (128 being standard). The environment has hosted two major competitions (IJCAI 2022 and NeurIPS 2022) in the past (it could be three, but the first one has almost no active participants). Some notable changes since version 1.x are:
- the complete rewrite of the code base,
- profession progression system and the market system,
- a simplified baseline (CleanRL),
- and a web client written by Parametrix.AI for the NeurIPS 2022 competition.

---
I have personally participated in the IJCAI 2022 and NeurIPS 2022 so I guess I could say that I am reasonably familiar with this environment. But I haven't worked with this environment since the last competition, therefore rating my confidence at 4.

---

> ### Author Response · Authors · 2023-08-09
> **Simple and Efficient Baselines - No Scripting!**
>
> Thank you for your thorough and extensive review of our work and for participating in our last competition. You are fully correct that 2.0 is a refinement of the original environment, rather than a fundamentally new project. The key scientific contribution is the flexible task system, enabling researchers to train on distributions of tasks and evaluate performance on objectives never seen during training. From an engineering perspective, implementing a task system requires a user-friendly mechanism for checking arbitrary conditions on game state. Coupled with improvements to efficiency, this took our team several iterations and months of work.
>
> You will be happy to know that the new baselines are purely neural, train in a day on 1 GPU, and support self-play via a flexible policy pool. We had to rewrite all of our middleware to accomplish this, which is under separate review as a standalone library, and this was by far the hardest engineering problem I have ever taken on. The original competition’s baseline was in RLlib, and almost nobody participated because of how buggy and difficult to use it was. I have personally taken calls with AnyScale to redesign multiple of their APIs and even offered to spend a few weeks on premise at their SF office to help fix their bugs -- for free. Unfortunately, RLlib has experienced several performance regressions from version to version, and everyone else I know has dropped it for now. I still have the utmost respect for the developers and what they are trying to accomplish, but for now, it is not stable or user friendly enough for us to use in our baselines.
>
> I completely understand your grievances with the previous competition. Yes, it is unfair to have to compete against industry labs. The problem is that, in order to regulate the amount of compute allowed, we would have to run your training scripts on our hardware (and allow you to make several submissions). That’s possibly 100k+ GPU hours and 1M+ CPU hours. Fortunately, we’ve secured funding for this, and in just a few days, you will be able to compete against everyone else with 8 A100 hours per submission. There’s still a track for industry to push the limits with lots of compute, but it’s a separate leaderboard.
>
> 2.0 includes an extra action that allows you to give items to teammates if they are standing on the same square. No more weird ID priority hacks.
>
> If the above improves your opinion of our work, please consider increasing your score. An accepted preprint would do wonders for participation in the competition and for progress on the platform.
>
> We would also like your opinion on the ordering of the sections of the paper, as one of the other reviewers suggested an alternative:
>
> Current ordering: Background, Novelty, Accessibility, Ethics, Neural MMO, Tasks, Performance, Limitations, Conclusion.
> Suggested order is: Novelty, Neural MMO, Background, Tasks, Performance, Limitations, Accessibility, Ethics, Conclusion.

---

> > ### Comment · Reviewer_3mZR · 2023-08-15
> > **Ordering of the sections of the paper**
> >
> > Hi,
> >
> > I see the point that Reviewer 3uba has made. The structure of the present manuscript does seem unconventional. People are used to the usual format of intro first and then related work. I think if the authors could re-organize in the way per Reviewer 3uba suggested, this could certainly enhance the reading experience for many.

---

> > > ### Author Response · Authors · 2023-08-19
> > > **Revision Uploaded**
> > >
> > > We have uploaded a revision with the new structure. Please let us know if there are any other improvements you would like to see in order to consider updating your score.

---

> > > > ### Comment · Reviewer_3mZR · 2023-08-19
> > > > **Thanks for the update**
> > > >
> > > > Thank you to the authors for the update. While I disagree with the authors' claim that this environment is easy to train without significant computational resources – as I believe the current MARL algorithms are not efficient enough for this, and considerable scripting would be involved – I must admit that the 2.0 update is a refinement of the original environment. The market and trading systems are intriguing, particularly as they are tied to character progression paths, opening up interesting research topics in the future. As the revisions have addressed most of my concerns, I have decided to raise my score.
> > > >
> > > > I hope that my previous experiences in the competition did not bias this review too much, and I hope the AC will consider my review accordingly.

---

> > > > > ### Author Response · Authors · 2023-08-21
> > > > > **Thank you**
> > > > >
> > > > > Thank you for revising your score. To be clear, it's definitely not *easy* to train, and additional compute and scripting both help, but we've found that you can get reasonable performance with simple network architectures. The larger problem was lack of infrastructure for handling hierarchical observations and actions as well as variable agent populations. At the least, our new baselines can provide a single-file training script derived from CleanRL that will train to reasonable performance on a 1 GPU desktop.

---

### Official Review · Reviewer_wNF1 · 2023-07-23
**a clean benchmark with a lot of users already**

**Rating:** 7
**Confidence:** 3
**Correctness:** The claims are correct as far as I ca…
**Clarity:** the paper is very clear.

**Strengths:**

The paper is well-written, the platform clearly represents a significant amount of work and is already successful

**Additional Feedback:**

There seems to have been extensive advertising for this work already -- e.g., through its use in the NeurIPS competition, which makes the incremental value of publishing the paper to the workshop smaller than other papers where the workshop is their major opportunity for publication.

**Documentation:**

There is extensive documentation and a clear maintenance plan for at least one year. There also is an active community on Discord.

**Ethics:**

I don't see any ethics problem assiociated to this submission.

**Limitations:**

The authors claim that the rules of their game are sufficiently far from reality that the platform cannot be used to train actual models of combat or commerce system. I think this is reasonable.

**Opportunities For Improvement:**

While I guess it is clear that the major scientific problems that are meant to be addressed relate to the complexity of tasks and multi-agent competition/collaboration, it is not very clear from the paper what this specific platform allows to unlock from a scientific perspective.

**Relation To Prior Work:**

The paper is clear on prior work regarding large-scale platforms for game-style reinforcement learning, less so on what these platforms have enabled in terms of research.

**Summary And Contributions:**

The paper describe Neural MMO 2.0, a platform designed to study reinforcement learning of large multi-agent problems (typically 128 agents). The platform has already been used in an accepted competition at NeurIPS, and is a rewrite+significant extension to previous versions. There already is an active community using this platform. The extensions involve the assively multi-agent setting, a new Web interface with visualization tools, as well as a rewrite of the code to make the engine faster, which leads to substancial speedups when running the environment is the bottleneck for learning.

---

> ### Author Response · Authors · 2023-08-09
> **Clarification of Scientific Impact**
>
> Thank you for your review of our work. We will formalize the below scientific contributions of our work and will include them in the camera ready if accepted:
>
> The vast majority of reinforcement learning research is currently conducted on simple, single-agent environments. The key problem with this is that e.g. Atari, Procgen cannot be used to evaluate many important properties of human cognition because they are not required to solve the environments. For example: can agents learn to cooperate with other learning agents? Can optimization methods learn policies that are stable in mixed competitive/cooperative settings? Can a single policy learn to accomplish a variety of different tasks? Can agents learn when other agents are changing the environment around them? Can agents create a realistic economy when given the ability to trade? These are all important features of human intelligence in the real world. Neural MMO provides a complex, cognitively rich environment in which to study these problems. At the same time, it is computationally accessible to academic researchers training on a single GPU. Previously, most work on these topics was limited to large industry groups with access to thousands of GPUs, i.e. XLand, OpenAI Five, AlphaStar.
>
> We would also like your opinion on the ordering of the sections of the paper, as one of the other reviewers suggested an alternative:
>
> Current ordering: Background, Novelty, Accessibility, Ethics, Neural MMO, Tasks, Performance, Limitations, Conclusion.
> Suggested order is: Novelty, Neural MMO, Background, Tasks, Performance, Limitations, Accessibility, Ethics, Conclusion.

---

### Official Review · Reviewer_3uba · 2023-07-27
**NeuralMMO 2.0: A High-Performance Rewrite with Enhanced Features and Usability**

**Rating:** 6
**Confidence:** 4
**Correctness:** See below.

**Strengths:**

This paper offers an in-depth examination of the Neural MMO 2.0 platform, elucidating its various components such as GameState, Predicates, and Tasks. Accompanied by detailed code snippets and comments, the paper effectively demonstrates the inner workings and structure of the platform, providing valuable insights for researchers and users alike.

**Additional Feedback:**

It is important to address the citation format errors and missing reference in the manuscript to ensure consistency and proper attribution. Please consider revising the following:

- L42 and L43: Correct the citation format according to the style guidelines.
- L59: Rectify the citation format error in accordance with the style guidelines.
- L147: Add an appropriate reference for "Tasks" to provide necessary context and acknowledge relevant work.

**Clarity:**

The paper's main issue is its lack of a clear structure, leading me to rate it a 3 for clear rejection. I suggest a complete rewrite of the paper to address this problem. Here are some points to consider, for example:

1. The paper has some structural issues regarding section order. I recommend reorganizing the sections as follows: 2 - 5 - 1 - 6 - 7 - 8 - 3 - 4 - 9. Sections 2 and 5 should serve as the introduction, while section 1 focuses on related work, and sections 6, 7, and 8 delve into the details. Sections 3 and 4 (Accessibility and Accountability/Ethics and Responsible Use) could be moved to the supplemental material, as they are more supplementary in nature. Finally, section 9 should be the conclusion.

2. The last two paragraphs of section 5 are disorganized. Instead of cramming all the differences between NMMO 1.0 and 2.0 into one paragraph with low-level details, it would be better to clearly list and discuss each point separately.

3. In section 6, the example of GameState should be moved to an appendix rather than being included in the main text. The detailed, low-level information is not user-friendly for newcomers who may lack context about NMMO. It is essential to remember that the paper is not an API documentation and should be written in a manner that is accessible and engaging for the target audience.

**Documentation:**

The NeuralMMO 2.0 project boasts extensive documentation on their website, ensuring users have access to all the necessary information. Additionally, the project includes a detailed maintenance plan, demonstrating the developers' commitment to continuous improvement and support. Moreover, NeuralMMO 2.0 is compatible with all operating systems, making it highly accessible for a wide range of users and increasing its applicability in various research contexts.

**Limitations:**

See below.

**Opportunities For Improvement:**

To enhance the experiment section, the authors should consider providing more RL agent baselines along with detailed descriptions of their settings. For instance, the authors could standardize several baseline environments within the NeuralMMO platform and conduct multiple experiments using widely-used algorithms like A2C and PPO. By presenting learning curves and sample efficiency metrics, the authors can effectively demonstrate the challenges involved in solving NeuralMMO tasks. This additional information will provide a more comprehensive understanding of the platform's capabilities and the complexities associated with training RL agents on NeuralMMO.

**Relation To Prior Work:**

Including a comparison table that highlights the key differences between NeuralMMO 1.0 and 2.0 would be extremely helpful for users and researchers. This table would provide a clear and concise overview of the improvements and new features introduced in version 2.0, facilitating a better understanding of the project's evolution and the advantages offered by the updated version.

**Summary And Contributions:**

This paper presents NeuralMMO 2.0, a comprehensive rewrite of the original NMMO 1.0. This updated version delivers a 3x speed improvement compared to its predecessor, offers a baseline RL experiment using CleanRL, and introduces a user-friendly web client, significantly enhancing the overall experience and functionality of the NeuralMMO platform.

---

> ### Author Response · Authors · 2023-08-09
> **PPO/A2C Baseline Now Available**
>
> Thank you for your time in reviewing our work and for your specific advice for improvements. We have provided a baseline and can reorder the sections. Please let us know if we can address any other concerns with the content or structure of our work in order to increase your opinion of the paper.
>
> A baseline model and compatible PPO/A2C implementation is now available as part of the starter kit for our NeurIPS 2023 competition, which is launching in a few days. From our experience running previous competitions, participants will quickly make large improvements. Instead of publishing an already outdated baseline in December, we plan to present it alongside the winning submissions to the competition. If you feel strongly that it would be better to include the baseline in this submission regardless, we would be happy to do so.
>
> We put Accessibility and Ethics early in the paper because the call for submissions emphasized these sections. However, we are open to your ordering if the other reviewers agree.
>
> We were unsure how to present material in the last paragraphs of section 5. To be clear, this does not mention any differences from Neural MMO 1.0 (this is in section 2). The section was instead meant to illustrate some of the complexity of what happens in the Neural MMO environment. Simply enumerating all of the features does not do the environment justice, and we were unsure how else to present the material in text form. If you do not find this section helpful, perhaps we could just link to https://neuralmmo.github.io/client for a demo?
>
> We will fix formatting errors and move the examples of GameState to an appendix if that is allowed. From the call, we were unsure if appendices could be included in the same document.

---

> > ### Author Response · Authors · 2023-08-19
> > **Restructured According to Your Suggestions**
> >
> > We have ordered the sections as per your suggestion, moved code snippets to an appendix, and removed the move experimental paragraphs in the Neural MMO section (the same information is available in a better suited wiki format on the documentation page). The new PDF is uploaded as a revision, and our PPO/A2C baseline is available here https://github.com/CarperAI/nmmo-baselines. If you feel that these changes address your concerns with the presentation of our work, kindly consider increasing your score. If not, please let us know what else we can improve.

---

> > > ### Comment · Reviewer_3uba · 2023-08-19
> > > **RE**
> > >
> > > Awesome! I've already changed the score accordingly. Thanks for updating the revision!

---

### Decision · Program_Chairs · 2023-09-22

**Decision:**

Accept (Poster)

**Comment:**

The paper describes Neural MMO 2.0, an improved environment for massive multiagent multitask RL experimentation.  In comparison to the previous version, this paper adds a flexible task system that enables research on generalization, open-endedness, and curriculum learning.  The technical advances are mostly engineering advances, but the framework has already been adopted by a large community and is enabling a concurrent competition at NeurIPS-2023.  The authors re-structured the paper based on the reviewers' comments.  This work is ready for publication.